# NuHepMC: A standardized event record format for neutrino event generators

**Steven Gardiner**[1][*]**, Joshua Isaacson**[1][†] **and Luke Pickering**[2][‡]

**1** Fermi National Accelerator Laboratory, P.O. Box 500, Batavia, IL 60510, USA
**2** STFC, Rutherford Appleton Laboratory, Harwell Oxford, United Kingdom

[*] gardiner@fnal.gov , [†] isaacson@fnal.gov , [‡] luke.pickering@stfc.ac.uk

## Abstract

Simulations of neutrino interactions are playing an increasingly important role in the pursuit of high-priority measurements for the field of particle physics. A significant technical barrier for efficient development of these simulations is the lack of a standard data format for representing individual neutrino scattering events. We propose and define such a universal format, named NuHepMC, as a common standard for the output of neutrino event generators. The NuHepMC format uses data structures and concepts from the HepMC3 event record library adopted by other subfields of high-energy physics. These are supplemented with an original set of conventions for generically representing neutrino interaction physics within the HepMC3 infrastructure.

# 1   Introduction

Worldwide experimental efforts in high-energy physics (HEP) are placing increasing emphasis on precision measurements of neutrinos. The pursuit of these measurements creates strong demands on the quality of neutrino event generators—the software tools used to simulate neutrino scattering in the context of experimental analyses [1, 2].

Recent discussions about the future of neutrino research, both at workshops focused on simulations of neutrino interactions [3] and as part of the HEP-wide Snowmass 2021 community planning process [4], have emphasized the need for greater flexibility in the use of neutrino event generators and related software. A particularly problematic technical barrier is the current lack of a common data format for representing the output *events*: lists of simulated particles involved in a neutrino interaction together with information describing their properties and relationships. At present, each neutrino event generator group maintains a unique output format, substantially complicating the use of multiple generators in large-scale experimental simulation workflows [5, 6]. The adoption of standard formats in the collider community has streamlined many components of the analysis pipeline. This has enabled straightforward analysis preservation in tools like Rivet [7, 8], allowed theorists to reinterpret experimental limits on one exotic physics scenario in light of others [9, 10], simplified comparisons between generators (see, e.g., Ref. [11]), and supported interoperability between simulation tools [12–14].

The only major software product that currently provides an official interface to the event formats produced by all four of the most widely-used neutrino event generators (GENIE [15, 16], GiBUU [17], NEUT [18, 19], and NuWro [20]) is NUISANCE [21], a framework for comparing simulation predictions to each other and to experimental data. Although this feature of NUISANCE has proven valuable for the field, its generic internal representation of the input events (the `FitEvent` C++ class) is too simplified for all applications, its event format conversion tools require linking to generator-specific shared libraries, and the need to support multiple evolving proprietary event formats represents a significant maintenance burden.

To facilitate further development of software products that interface with neutrino event generators, as well as to support the use of a wider variety of simulation-based neutrino interaction models in experimental analyses, we present a new event format, *NuHepMC*, as a universal standard to be adopted by consensus of the neutrino event generator community. The data structures, file formats, and basic concepts in NuHepMC are identical to the versatile and mature HepMC3 library [22] adopted by other subfields of HEP. Using HepMC3 as a foundation, we define in NuHepMC an extensible and extendable set of conventions for representing neutrino interaction physics in a tool-agnostic way. This approach provides a generic event format that can be used in all future simulation development for neutrino experiments. Because we avoid placing any limitations on the information that individual event generators may output, the NuHepMC standard enables lossless, bidirectional conversion between HepMC3 and the existing proprietary neutrino event formats.

In Sec. 2, we present the specification of the NuHepMC format. The flux-averaged total cross section, a particularly important quantity for analyzing neutrino scattering simulations, is discussed in Sec. 3. Finally, Sec. 4 uses NuHepMC as an output format for GENIE, NuWro, NEUT, ACHILLES [23], and MARLEY [24] as well as an input format in NUISANCE to demonstrate a first NuHepMC-based analysis of neutrino event generator predictions.

## 2 Specification

The details of the NuHepMC specification are broken down into three categories that describe four components from HepMC3. Each element of the specification is labeled as a *Requirement*, a *Convention*, or a *Suggestion*. The requirements dictate a minimum level of information to be included when writing out events. The conventions are optional details that an event generator group can choose to include or omit while still conforming to the NuHepMC standard. Finally, the suggestions are optional recommendations that are less strongly encouraged than the conventions.

The four HepMC3 components considered in this standard are the generator run metadata, event metadata, vertex information, and particle information. Specifications for each of these components can be found in Secs. 2.3, 2.4, 2.5, and 2.6 respectively.

### 2.1 Labeling scheme

The elements of the specification are enumerated below in the form <Component>.<Category>.<Index>, where the component of interest is given as G for generator run metadata, E for event metadata, V for vertex information, and P for particle information. The category is denoted by R for a requirement, C for a convention, and S for a suggestion. For example, the second convention for event metadata would be labeled as E.C.2.

If conventions or suggestions prove useful, become widely adopted, and are considered stable, they may become requirements in future versions of this specification. G.C.1 defines a convention for event generator authors to signal whether or not specific optional elements of this specification have been followed.

### 2.2 HepMC3 C++ classes

Throughout this standard, references are made to various HepMC3 C++ classes, e.g., `HepMC3::GenRunInfo`. However, these are used as a convenient handle for data objects. This specification should not be considered specific to the HepMC3 C++ reference implementation.

### 2.3 Generator Run Metadata

The generator run metadata describes the overall setup of the event generator, i.e., information that is not unique to a specific event. The NuHepMC specifications for this metadata are as follows:

G.R.1 VALID GENRUNINFO:
All NuHepMC Vectors must contain a `HepMC3::GenRunInfo` instance.

G.R.2 NUHEPMC VERSION:
A NuHepMC `HepMC3::GenRunInfo` instance must contain the following attributes that specify the version of NuHepMC that is implemented:

- type: `HepMC3::IntAttribute`,
  name: `"NuHepMC.Version.Major"`

- type: HepMC3::IntAttribute,
  name: "NuHepMC.Version.Minor"

- type: HepMC3::IntAttribute,
  name: "NuHepMC.Version.Patch"

This document describes **version 0.9.0** of NuHepMC.

G.R.3 GENERATOR IDENTIFICATION:
A NuHepMC HepMC3::GenRunInfo instance must contain a
HepMC3::GenRunInfo::ToolInfo for each 'tool' involved in the production of
the Vector thus far. The ToolInfo instance must contain non-empty name and
version fields.

G.R.4 PROCESS METADATA:
A NuHepMC HepMC3::GenRunInfo instance must contain a HepMC3::VectorIntAttribute
named "NuHepMC.ProcessIDs" listing all physics process IDs as integers. For
each valid process ID, the HepMC3::GenRunInfo instance must also contain two
other attributes giving a name and description of each:

- type: HepMC3::StringAttribute,
  name: "NuHepMC.ProcessInfo[<ID>].Name"

- type: HepMC3::StringAttribute,
  name: "NuHepMC.ProcessInfo[<ID>].Description"

where <ID> enumerates all process IDs present in "NuHepMC.ProcessIDs".
(See also E.C.1).

G.R.5 VERTEX STATUS METADATA:
The NuHepMC HepMC3::GenRunInfo instance must contain a
HepMC3::VectorIntAttribute named "NuHepMC.VertexStatusIDs" declar-
ing any generator-specific status codes used. Including the standard HepMC3
codes in this list is optional, but they must not be reused to mean something
different than in the HepMC3 specification. For each declared vertex status, the
HepMC3::GenRunInfo instance must also contain two other attributes giving a
name and description of each:

- type: HepMC3::StringAttribute,
  name: "NuHepMC.VertexStatusInfo[<ID>].Name"

- type: HepMC3::StringAttribute,
  name: "NuHepMC.VertexStatusInfo[<ID>].Description"

where <ID> enumerates all status codes present in "NuHepMC.VertexStatusIDs"
(See also V.R.1).

G.R.6 PARTICLE STATUS METADATA:
The NuHepMC HepMC3::GenRunInfo instance must contain a
HepMC3::VectorIntAttribute named "NuHepMC.ParticleStatusIDs" declar-
ing any generator-specific status codes used. Including the standard HepMC3
codes in this list is optional, but they must not be reused to mean something
different than in the HepMC3 specification. For each valid particle status, the
HepMC3::GenRunInfo instance must also contain two other attributes giving a
name and description of each:

- type: `HepMC3::StringAttribute`,
  name: `"NuHepMC.ParticleStatusInfo[<ID>].Name"`

- type: `HepMC3::StringAttribute`,
  name: `"NuHepMC.ParticleStatusInfo[<ID>].Description"`

where <ID> enumerates all status codes present in `"NuHepMC.ParticleStatusIDs"` (see P.R.1 for more details).

G.R.7 EVENT WEIGHTS:
For weights that will be calculated for every event, HepMC3 provides an interface for storing the weight names only once in the `HepMC3::GenRunInfo` instance. At least one event weight named `"CV"` must be declared in the `HepMC3::GenRunInfo` instance and filled for every event.

This weight may be 1 or constant for every event in a generator run (in the case of an *unweighted* event vector). This weight must always be included by a user when producing correctly-normalized predictions from a NuHepMC Vector and must not be assumed to be always 1. The exact form of this weight and whether it is the only information required to properly normalize a prediction are considered implementation details.

G.R.8 NON-STANDARD PARTICLE NUMBERS (PDG CODES):
Essentially all event generators in HEP use a standard set of integer codes to identify particle species. This numbering scheme is maintained by the Particle Data Group (PDG) and is regularly updated in their Review of Particle Physics [25, Sec. 45, p. 733].

We expect that neutrino event generators may need to use codes for non-standard particle species (i.e., those without an existing PDG code) for a variety of applications. This could include simulating exotic physics processes involving new particles as well as implementing bookkeeping methods involving generator-specific quasiparticles.

The NuHepMC `HepMC3::GenRunInfo` instance must contain a `HepMC3::VectorIntAttribute` named `"NuHepMC.AdditionalParticleNumbers"` declaring any particle codes used that are not defined in the current PDG numbering scheme. Including any of the standard codes in this list is permitted but not required. The standard particle codes must not be reused to mean something different than in the PDG specification.

For each additional particle code, the `HepMC3::GenRunInfo` instance must also contain an attribute giving a unique name to the represented particle species:

- type: `HepMC3::StringAttribute`,
  name: `"NuHepMC.AdditionalParticleInfo[<PDG>].Name"`

where <PDG> enumerates all particle numbers present in `"NuHepMC.AdditionalParticleNumbers"`.

See also G.C.8 for a suggested way of storing descriptions of these special particle species.

G.C.1 SIGNALING FOLLOWED CONVENTIONS:
To signal to a user that an implementation follows a named convention from this specification, a `HepMC3::VectorStringAttribute` should be added to the

HepMC3::GenRunInfo instance named "NuHepMC.Conventions" containing the names of the conventions adhered to.

G.C.2 VECTOR EXPOSURE (STANDALONE):
Each Vector should contain a description of the exposure of the generator run. When running a standalone event simulation this will often correspond to the number of events requested, which may differ from the number of events output in cases where events that are not written out must contribute to the total cross section calculation.

- type: HepMC3::LongAttribute,
  name: "NuHepMC.Exposure.NEvents"

Implementations should not adhere to both G.C.2 and G.C.3 simultaneously.

G.C.3 VECTOR EXPOSURE (EXPERIMENTAL):
Each Vector should contain a description of the exposure of the generator run. When simulating with some experimental exposure, often represented for accelerator neutrino experiments in units of "protons on target" (POT), the exposure should be described. Two attributes are reserved for signaling the exposure used to users. One or both can be provided.

- type: HepMC3::DoubleAttribute, name: "NuHepMC.Exposure.POT"
- type: HepMC3::DoubleAttribute, name: "NuHepMC.Exposure.Livetime"

Implementations should not adhere to both G.C.2 and G.C.3 simultaneously.

G.C.4 CROSS SECTION UNITS AND TARGET SCALING:
There are a variety of units typically used to report both measured and predicted cross sections in HEP. For neutrino cross sections specifically, $10^{-38}$ cm$^2$ per nucleon is common, but not ubiquitous. We want to provide a sensible recommended default while preserving the flexibility for an implementation to signal a different choice. One or both of the following HepMC3::StringAttributes may be included on the HepMC3::GenRunInfo to indicate the cross section units used within a vector.

- "NuHepMC.Units.CrossSection.Unit". Possible values of the attribute are not restricted, but we reserve the meanings of the following:
  - "pb": Picobarns or $10^{-36}$ cm$^2$. This is our recommended default.
  - "cm2": Using bare cm$^2$ in this option, without any power-of-ten scaling, is not recommended due to numerical precision concerns. The natural scale of neutrino–nucleon cross sections is approximately $10^{-38}$, which is very close to the minimum representable IEEE 754 single-precision floating point number [26].
  - "1e-38 cm2": The choice of $10^{-38}$ cm$^2$ in this option is the most frequent in the neutrino literature.
- "NuHepMC.Units.CrossSection.TargetScale". Possible values of the attribute are not restricted, but we reserve the meanings of the following for ease of compatibility with existing conventions:
  - "PerTargetAtom": Our recommendation. Choosing "atom" rather than "nucleus" in this context removes ambiguity when considering neutrino interactions with atomic electrons.

– `"PerTargetMolecule"`: Sometimes used for hydrocarbon- and water-target measurements.

– `"PerTargetNucleon"`: A common choice in the literature.

– `"PerTargetMolecularNucleon"`: Another common choice in the literature and in existing neutrino event generators. We recommend against implementations using this scheme.

If this convention is signalled, the chosen units should be assumed to apply to cross section information stored according to G.C.5, E.C.4, E.C.2, and E.C.3. For further discussion of the schemes detailed above and the motivation for recommending against `"PerTargetMolecularNucleon"`, see A.

It is ultimately up to the user to parse these attributes and decide whether any additional scaling is needed for their purposes. If these attributes are not present, then the cross section should be assumed to be in picobarns per target atom. We *strongly* recommend that implementations use this default.

G.C.5 FLUX-AVERAGED TOTAL CROSS SECTION:
The flux-averaged total cross section, $\langle \sigma \rangle$, is a scaling factor that is needed to convert a distribution of simulated events into a prediction of the flux-averaged cross-section—an experimentally-accessible quantity. Details on the definition of this quantity are given in Sec. 3.1.

The value of $\langle \sigma \rangle$ is not always known at the beginning of a generator run. As described in B, a running estimate of the flux-averaged total cross section may be computed as events are generated. Element E.C.4 of this specification provides a means of storing the value of this running estimate in each event. If known at the start of a run, the value of $\langle \sigma \rangle$ should be stored as a `HepMC3::DoubleAttribute` in the `HepMC3::GenRunInfo` named `"NuHepMC.FluxAveragedTotalCrossSection"`.

G.C.6 CITATION METADATA:
Modelling components implemented based on published work should always be fully cited. The `HepMC3::GenRunInfo` should contain at least one `HepMC3::VectorStringAttribute` for each relevant modelling component, named according to the pattern `"NuHepMC.Citations.<Comp>.<Type>"`. Valid substitutions for the <Comp> and <Type> fields are not restricted by this standard beyond the requirement that they are pure mixed-case alpha-numeric. We suggest using <Comp>=`Generator` for specifying the main citation for the interaction generator and <Comp>=`Process[<ID>]` for individual processes. For common reference formats in the HEP field, we suggest some common values for the <Type> field:

- `"InspireHEP"` might contain one or more unique InspireHep identifiers (texkeys).

- `"arXiv"` might contain one or more unique arXiv identifiers (eprint numbers).

- `"DOI"` might contain one or more unique Digital Object Identifiers.

We hope that automatic bibliography generation tools using this metadata will be built.

G.C.7 BEAM ENERGY DISTRIBUTION DESCRIPTION:
Each vector should contain a description of the beam particle flux used to sim-

ulate the output event vector. For many truth studies and experimental simulations where the detector is not physically close to the source, a simple beam energy distribution is enough to describe the particle beam. The two types of energy distribution covered by this convention are mono-energetic beams and those with distributions described by a simple histogram. The type should be signalled via a `HepMC3::StringAttribute` named `"NuHepMC.Beam.Type"` with value `"MonoEnergetic"` or `"Histogram"` stored on the `HepMC3::GenRunInfo`. For both types, relevant units can be signalled via two attributes:

- `"NuHepMC.Beam.EnergyUnit"`. Possible values of the attribute are not restricted, but we reserve the meanings of `"MEV"` and `"GEV"`. This attribute should always exist and be not empty.

- `"NuHepMC.Beam.RateUnit"`. Possible values of the attribute are not restricted, but we reserve the meaning of `"Arbitrary"` to signal that the normalization of the distribution was not known or used by the simulation. If this attribute is not used then the normalization will be assumed arbitrary.

For the case of a `"MonoEnergetic"`-type distribution, all beam particles in the vector must have identical energy. The attribute `"NuHepMC.Beam[<PDG>].MonoEnergetic.Energy"` can be used to signal the beam energy in the lab frame, but the usage of this attribute is optional as the energy can be determined from the first (or any) event in the vector.

For the case of a `"Histogram"`-type distribution, the histogram should be encoded into two `HepMC3::VectorDoubleAttribute` per beam species on the `HepMC3::GenRunInfo`:

- `"NuHepMC.Beam[<PDG>].Histogram.BinEdges"`
- `"NuHepMC.Beam[<PDG>].Histogram.BinContent"`

where <PDG> enumerates the PDG particle numbers of all beam particles present in the event vector. *N.B.* the number of entries in the `"BinEdges"` vector should always be one more than the number of entries in the `"BinContent"` vector.

The `HepMC3::BoolAttribute`,

- `"NuHepMC.Beam[<PDG>].Histogram.ContentIsPerWidth"`,

should be used to signal that the number of neutrinos in a given histogram is found by multiplying the bin content by the bin width, rather than from the content alone. While this might be determined by parsing the `RateUnit` attribute, existing neutrino generators make different assumptions when sampling input neutrino beam energy distributions, so we specify an explicit attribute. If this attribute is not provided, then it is expected that the number of neutrinos in a given bin is specified by the bin content alone and is independent of the width of the bin.

For a suggestion on how to encode useful information about more realistic neutrino beam descriptions, see E.S.1.

G.C.8 NON-STANDARD PARTICLE NUMBER DESCRIPTIONS:
For each additional particle number <PDG> declared in the `"NuHepMC.AdditionalParticleNumbers"` attribute, according to G.R.8, the `HepMC3::GenRunInfo` instance may contain an attribute giving a description of the particle:

- type: HepMC3::StringAttribute,
  name: "NuHepMC.AdditionalParticleInfo[<PDG>].Description"

For non-standard particles that should be further simulated by particle propagation simulations, such as GEANT4 [27], additional information encoded here may be enough to enable automatic propagation. In this version of NuHepMC, we do not attempt to prescribe a format for such information but highlight that HepMC3::GenRunInfo attributes of the form,
"NuHepMC.AdditionalParticleNumber[<PDG>].<SimName>.<AttrName>",
might be a useful for communicating such additional information. These additional attributes should include, at a minimum, the particle's mass, width, spin, and electric charge.

G.S.1 RUN CONFIGURATION:
It is suggested that a NuHepMC HepMC3::GenRunInfo instance should contain all information required to reproduce the events in the vector. This may be stored in attributes with names beginning with "NuHepMC.Provenance". The information required will necessarily be generator-specific, but we suggest two attributes that would be helpful to downstream users:

- type: HepMC3::LongAttribute,
  name: "NuHepMC.Provenance.NEvents
- type: Implementation defined,
  name: "NuHepMC.Provenance.RNGState"
  - This might be a single number used as the seed to initialize the random number generator (RNG). It could also be a more complicated description of the RNG state.

G.S.2 COMPLETE STATUS METADATA:
While G.R.5 and G.R.6 explicitly do not require implementations to emit metadata for standard status codes defined in the HepMC3 standard, it is suggested that the complete list of status codes used by an implementation are included in the "NuHepMC.VertexStatusInfo" and "NuHepMC.ParticleStatusInfo" attributes.

## 2.4 Event Data

The event is used to store information about the event as a whole. An event is described by arbitrary metadata and a graph of particles (edges) and vertices (nodes), each with their own arbitrary metadata. The NuHepMC specifications for events are as follows:

E.R.1 HEPMC3 COMPATIBILITY:
The HepMC3 standard places some constraints on valid event graphs, these constraints must be respected by valid NuHepMC events as we require full compatibility with HepMC3. More details of these constraints can be found in Ref. [22].

Existing neutrino event generators often rely on effective descriptions of the nuclear environment in a neutrino–nucleus hard scattering process. This means that four-momentum is often not explicitly conserved for the neutrino–nucleus system. Energy and momenta can be exchanged with a *nuclear remnant*, which is not directly involved in a neutrino–nucleon hard scatter, through initial and final state interactions. Implementations are free to conserve four momentum and emit all physical initial and final state particles, including the fully-simulated

nuclear remnant, but for those implementations where such a requirement is not feasible or would delay the adoption of this standard, P.C.2 reserves a non-standard particle number that can be used to represent a nuclear remnant that is not precisely simulated.

E.R.2 EVENT NUMBER:
Each HepMC3::GenEvent must have a non-negative event number that is unique within a given vector.

E.R.3 PROCESS ID:
The process ID for the primary physics process that is represented in the HepMC3::GenEvent must be recorded in a HepMC3::IntAttribute named "signal_process_id". The metadata defining this process ID must be stored according to G.R.4.

E.R.4 UNITS:
Energy and position units must be explicitly set in the HepMC3::GenEvent.

E.R.5 LAB POSITION:
The position of the event in the lab frame must be added as a HepMC3::VectorDoubleAttribute, named "LabPos", with the same units as used when implementing E.R.4. See E.C.5 for how to optionally store time in this attribute. If the simulation did not involve a macroscopic geometry, then this variable may be set to [0, 0, 0].

E.R.6 VERTICES:
An event must contain at least one HepMC3::GenVertex, and must have one and only one with a *primary interaction vertex* status code. No HepMC3::GenVertex may have a *not defined* status code. (See V.R.1 for additional details).

E.R.7 BEAM AND TARGET PARTICLES:
An event must contain exactly one particle with the *incoming beam particle* status code and one particle with the *target particle* status code (see P.R.1). We recommend that, in cases where the colliding initial-state particles are distinct, the more massive of the two should be considered the target. For neutrino scattering, the target will thus often be a complex nucleus or a free nucleon. In the case of equally massive particles, the choice to label one of them as the target is arbitrary.

P.C.1 provides a convention for marking a constituent bound nucleon struck by the incoming beam particle in the event graph.

E.R.8 EVENT COMPLETENESS:
All simulated incoming and outgoing physical particles must be written to the event. The storage of intermediate particles is considered an implementation detail.

E.C.1 PROCESS IDS:
It is not appropriate to mandate a specific set of interaction processes and assign them IDs in this standard. Different models make different choices, and it is impossible to foresee modeling developments that would require new process IDs to be defined in the future. Instead, the ranges of IDs given below are recommended for high-level categorization of processes. Even if an implementation uses the convention in Table 1, it must still adhere to G.R.4.

Charged-current (CC) processes should have identifiers in the X00-X49 block, and neutral-current (NC) processes should have them in the X50-X99 block. Negative

| Identifier | Process |
|:---:|:---:|
| 100-199 | Low-Energy Nuclear Scattering |
| 200-299 | Quasielastic |
| 300-399 | Meson Exchange Current |
| 400-499 | Resonance production |
| 500-599 | Shallow inelastic scattering |
| 600-699 | Deep inelastic scattering |
| 700- | Other process types |

Table 1: Process ID ranges for various process categories.

process IDs may be reserved for electromagnetic interactions in neutrino event generators that include them.

E.C.2 TOTAL CROSS SECTION:
The total cross-section for the incoming beam particle, with its specific energy, to interact with the target particle should be stored in a `HepMC3::DoubleAttribute` on the `HepMC3::GenEvent`, named `"TotXS"`. See G.C.4 for conventions on signalling cross section units.

E.C.3 PROCESS CROSS SECTION:
The total cross-section for the selected process ID for the incoming beam particle, with its specific energy, to interact with the target particle should be stored in a `HepMC3::DoubleAttribute` on the `HepMC3::GenEvent`, named `"ProcXS"`. See G.C.4 for conventions on signalling cross section units.

E.C.4 ESTIMATED FLUX-AVERAGED TOTAL CROSS SECTION:
Some simulations build up an estimate of the flux-averaged total cross section $\langle \sigma \rangle$ as they run. This makes implementing G.C.5 impractical in many cases. As an alternative, the built-in attribute `HepMC3::GenCrossSection`, accessed via `GenEvent::cross_section` should be used to store the current estimate of $\langle \sigma \rangle$. A user can then use the best estimate provided with the last generated event to correctly scale an event rate to a cross-section prediction.

For event generators that do not currently provide the value of $\langle \sigma \rangle$ in the output, B provides suggestions for algorithms for computing a running estimate and associated Monte Carlo statistical uncertainty as events are produced.

When implementing this convention, ensure that the `cross_sections` and `cross_section_errors` data members are the same length as the number of weights defined in the header. These should be filled with the current estimate of the total cross section for each variation based on all events generated so far, including the current event. Additionally, the `HepMC3::GenCrossSection` data members `accepted_events` and `attempted_events` should be filled with appropriate values.

E.C.5 LAB TIME:
If the `"LabPos"` attribute vector contains three entries then it is assumed to only contain the spatial position of the event. If it contains four entries, then the fourth entry is interpreted as the time of the event in seconds.

E.S.1 BEAM DESCRIPTION (BEAM SIMULATION)
For more complex beam simulations that can not adequately be described by a

single energy or energy histogram (see G.C.7), it is suggested that the full parent decay history is included in the `HepMC3::GenEvent`. A full set of conventions for the description of beam particle production and parent particle decay chains (for the case of neutrino beams) is currently outside the scope of this specification, but generator implementations can signal that they adhere to this suggestion to notify users that some or all of the beam particle production information is included in the event.

## 2.5 Vertex Information

The vertices in a HepMC3 event are used to connect groups of incoming and outgoing particles. For the vertex information, there is only one requirement in the present version of the NuHepMC standard.

V.R.1 VERTEX STATUS CODES:
We extend the HepMC3 definition of `HepMC3::GenVertex::status` to include the concept of a primary vertex, corresponding to the *primary* process (i.e., the one labelled according to E.C.1), and a final state interaction (FSI) summary vertex. The full set of defined status codes can be found in Table 2. Implementations are free to define specific vertex status codes to refer to individual FSI (or ISI) processes and output as much information as they require. However, a single summary vertex may be useful for some purposes if the full FSI history is very detailed or not often needed by users.

| Status Code | Meaning | Usage |
|---|---|---|
| 0 | Not defined | Do not use |
| 1 | Primary interaction vertex | Recommended for all cases |
| 2 | FSI Summary vertex | Recommended for all cases |
| 3-20 | Reserved for future NuHepMC standards | Do not use |
| 21-999 | Generator-dependent | For generator usage |

Table 2: Set of vertex status codes

Any secondary vertex included within a NuHepMC event may have a status between 21 and 999. Note that G.R.5 requires that all generator-specific status codes must be fully described by attributes stored in the `HepMC3::GenRunInfo`.

V.C.1 BOUND NUCLEON SEPARATION VERTEX
When an interaction with a nucleon bound within a nucleus with definite kinematics is simulated, a `HepMC3::GenVertex` corresponding to the separation of the struck nucleon and the nuclear remnant may be included and assigned status code 21. If this convention is signalled via the mechanism described in G.C.1, then status code 21 need not be included in the implementation of G.R.5.

## 2.6 Particle Information

In the current version of the NuHepMC standard, there is only a single requirement and two conventions for the particle information.

P.R.1 PARTICLE STATUS CODES:
We extend the HepMC3 definition of `HepMC3::GenParticle::status` slightly

| Status Code | Description | Usage |
|:---:|:---:|:---:|
| 0 | Not defined | Do not use |
| 1 | Undecayed physical particle | Recommended for all cases |
| 2 | Decayed physical particle | Recommended for all cases |
| 3 | Documentation line | Used for in/out particles in the primary process |
| 4 | Incoming beam particle | Recommended for all cases |
| 5-10 | Reserved for future HepMC3 standards | Do not use |
| 11-19 | Reserved for future NuHepMC standards | Do not use |
| 20 | Target particle | Recommended for all cases |
| 21-200 | Generator-dependent | For generator usage |
| 201- | Simulation-dependent | For simulation software usage |

Table 3: Particle status codes

to include the concept of a target particle. For neutrino scattering, this will usually be a target nucleus. The status codes are defined in Table 3.

Note especially that any incoming real particle must have a status code of 4 or 20, and any outgoing real particle must have a status code of 1. This allows users to know at a glance which simulated particles must be considered "observable" and which are "internal" details of the calculation. Special care must be taken when including the effects of initial-state and final-state interactions.

Any internal particle included within a NuHepMC event may have a status in the range than 21-200. Note that G.R.6 requires that all generator-specific status codes must be fully described by attributes stored in the `HepMC3::GenRunInfo`.

P.C.1 PARTICLE STATUS CODES:
When an interaction with a bound nucleon with definite kinematics is simulated, the internal `HepMC3::GenParticle` corresponding to the bound nucleon should have status code 21. If this convention is signalled via the mechanism described in G.C.1, then status code 21 need not be included in the implementation of G.R.6.

P.C.2 NUCLEAR REMNANT PARTICLE CODE:
HepMC3 places restrictions on all external particles in the event graph to facilitate automatic checking of four momentum conservation at the event graph level. As a result, we define the new particle number 2009900000 to correspond to a nuclear remnant pseudo-particle. This particle corresponds to an implementation detail that should be used to abide by HepMC3 constraints to not have external vertices, but should not be considered for physics analyses or onward simulation. The number is chosen, according to the PDG scheme [25, Sec. 45, p. 733], to be outside the range reserved for nuclear and quark-content particles and signals that it is a non-standard code by having the 6th and 7th least significant digits set to 9.

If this convention is signalled via the mechanism described in G.C.1, then particle number 2009900000 need not be included in the implementation of G.R.8.

If the nuclear particle number of the remnant is known, it can be added as a `HepMC3::IntAttribute` on the `HepMC3::GenParticle` named `"remnant_particle_number"`.

## 3  Flux-averaged total cross section

Comparisons of event generator predictions to external model calculations and experimental data typically involve the conversion of simulated event distributions to total or differential cross sections. This conversion is usually made using a scaling factor $\langle \sigma \rangle$ called the flux-averaged total cross section. This factor is simple to use but often difficult to calculate analytically. Given the importance of $\langle \sigma \rangle$ for analyses of simulated neutrino scattering events, we provide mathematical details about its definition and calculation.

Section 3.1 derives an expression for $\langle \sigma \rangle$ that is generally applicable to most simulations of interest for neutrino experiments, including those with a time-dependent neutrino source and a full treatment of the detector geometry. Example methods for obtaining Monte Carlo estimators of $\langle \sigma \rangle$ suitable for storage via E.C.4 are described in B. Section 3.2 describes some simple cases in which $\langle \sigma \rangle$ may be calculated directly, making them good candidates for situations in which event generators may implement G.C.5.

### 3.1  Derivation

Let $\phi(f, E_\nu, \vartheta_\nu, \varphi_\nu, \mathbf{x}, t)$ be the differential flux of (anti)neutrinos of species $f$ with energy $E_\nu$, momentum direction (expressed in terms of the Cartesian unit vectors)

$$\hat{\mathbf{p}}_\nu = \sin\vartheta_\nu \cos\varphi_\nu \hat{\mathbf{x}} + \sin\vartheta_\nu \sin\varphi_\nu \hat{\mathbf{y}} + \cos\vartheta_\nu \hat{\mathbf{z}}, \tag{1}$$

and instantaneous three-position $\mathbf{x}$ at time $t$. This quantity is defined so that

$$\Phi(\mathbf{x}) = \sum_f \int dE_\nu \, d\cos\vartheta_\nu \, d\varphi_\nu \, dt \, \phi \tag{2}$$

has units of integrated flux (e.g., $\mathrm{cm}^{-2}$).

Assuming that the relevant neutrino interaction cross sections are sufficiently small that beam attenuation and multiple scattering effects can be neglected, then the total number of interactions $N$ expected in a volume of interest

$$V = \int d^3\mathbf{x} \tag{3}$$

is given by

$$N = \sum_f \sum_j \int dE_\nu \, d\cos\vartheta_\nu \, d\varphi_\nu \, d^3\mathbf{x} \, dt \, \phi \, \rho \, \sigma \tag{4}$$

where $\rho = \rho(j, \mathbf{x})$ is the number density of the $j$-th kind of target at position $\mathbf{x}$. The symbol $\sigma = \sigma(f, j, E_\nu)$ denotes the total cross section for (anti)neutrinos of species $f$ and energy $E_\nu$ to interact with the $j$-th kind of target (typically a particular nuclide).

One may rewrite the expression for $N$ in the simple form

$$N = \langle \Phi \rangle \cdot \langle T \rangle \cdot \langle \sigma \rangle \tag{5}$$

where

$$\langle \Phi \rangle = \frac{1}{V} \sum_f \int dE_\nu \, d\cos\vartheta_\nu \, d\varphi_\nu \, d^3\mathbf{x} \, dt \, \phi \tag{6}$$

is the average flux in the volume $V$,

$$\langle T \rangle = \frac{1}{\langle \Phi \rangle} \sum_f \sum_j \int dE_\nu \, d\cos\vartheta_\nu \, d\varphi_\nu \, d^3\mathbf{x} \, dt \, \phi \, \rho \tag{7}$$

is the flux-averaged number of targets illuminated by the (anti)neutrinos, and

$$\langle \sigma \rangle = \frac{N}{\langle \Phi \rangle \cdot \langle T \rangle} \tag{8}$$

is the flux-averaged total cross section. For a discussion on how these might be calculated using Monte-Carlo methods see B.

## 3.2 Analytic calculation under simplifying assumptions

In certain simple cases, the flux-averaged total cross section $\langle \sigma \rangle$ is known at the start of the run and can be computed analytically to implement G.C.5. A common example is the case of a point target of type $b$ that is exposed to uni-directional beam of (anti)neutrinos of species $a$ with a fixed emission time. In this case, we may write the differential flux and target density as

$$\phi = A(E_\nu)\,\delta(\cos\vartheta_\nu - \cos\theta_0)\,\delta(\varphi_\nu - \phi_0)\,\delta(t - t_0)\,\delta_{fa} \tag{9}$$

$$\rho = B\,\delta^3(\mathbf{x} - \mathbf{x}_0)\,\delta_{jb}\,. \tag{10}$$

Here $\delta_{xy}$ is the Kronecker delta and the constants $\theta_0$, $\phi_0$, $t_0$, $\mathbf{x}_0$, $a$, and $b$ are arbitrary values of the relevant variables. The function $A(E_\nu)$ gives the incident (anti)neutrino energy spectrum (with unimportant normalization) and has units of flux divided by energy (e.g., $\text{GeV}^{-1}\,\text{cm}^{-2}$). The constant $B$ is dimensionless. From Eqs. 6, 7, and 8, it follows that

$$\langle \Phi \rangle = \int dE_\nu A(E_\nu) \tag{11}$$

$$\langle T \rangle = B \tag{12}$$

$$\langle \sigma \rangle = \frac{\int dE_\nu A(E_\nu)\,\sigma(a,b,E_\nu)}{\int dE_\nu A(E_\nu)}\,. \tag{13}$$

If the function $A(E_\nu)$ and the total cross section $\sigma(a,b,E_\nu)$ are accessible during generator initialization, then the integrals in Eq. 13 are easily computable by standard numerical methods.

In the even simpler case where the neutrino source is also monoenergetic, then $A(E_\nu)$ contains a Dirac delta function, and the flux-averaged total cross section is just the total cross section evaluated at the fixed energy of the beam.

# 4 NuHepMC Generator Predictions

Using preliminary tools for converting proprietary neutrino event formats to the NuHepMC standard defined above, we demonstrate the utility of the common format with some cross-section predictions.

A comparison to a recent neutrino-argon cross-section measurement from the MicroBooNE collaboration [28] is shown on the left-hand side of Fig. 1. This comparison was made with the NUISANCE framework, which before this implementation of NuHepMC would have to have been built against GENIE, NEUT, and NuWro binaries of compatible versions to be able to generate the predictions shown in the figure. In the present workflow, results can be obtained using only generator-agnostic tools for interpreting events stored in the NuHepMC format. This relatively small reduction of technical requirements will dramatically lower the barrier to making high-quality prediction-data comparisons for many experts and non-experts in the neutrino community.

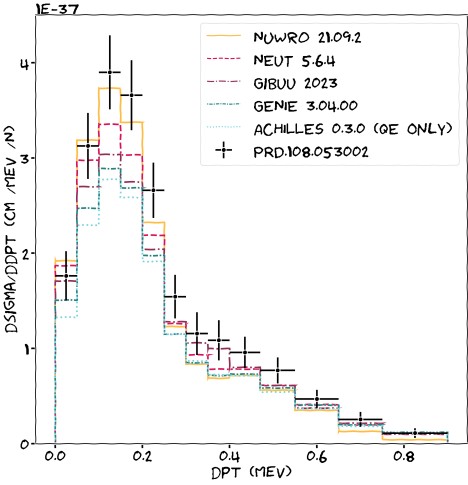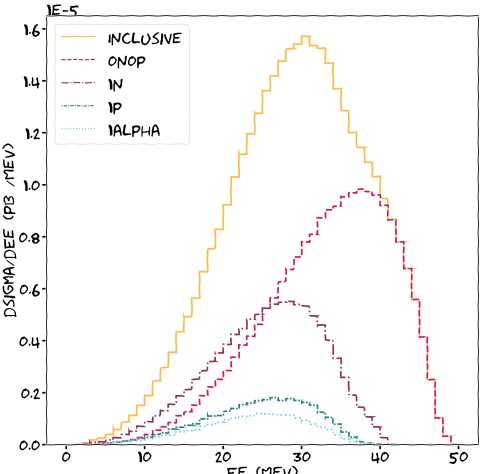

Figure 1: (*left*) Comparison of NuWro, NEUT, GiBUU, GENIE, and ACHILLES to the $\delta p_T$ data from the MicroBooNE collaboration [28]. The comparison is a demonstration of the event generation to NUISANCE pipeline and not a physics statement about the prediction quality of any generator. (*right*) Low-energy differential cross section predictions produced using MARLEY events in the NuHepMC format.

It is particularly worth noting that the different generator predictions in this comparison provide critical cross-section scaling information in different ways. Thanks to the provisions in this standard, NUISANCE is able to use G.C.1 to determine what scaling methods are available and then automatically scale the selected event rate to a cross-section measurement prediction without any user input or generator-specific code or knowledge. For reference, the NEUT and NuWro selected event projections are scaled via G.C.5, while GiBUU, GENIE, and Achilles are scaled via E.C.4.

We want to emphasise that the comparison to data is solely a demonstration of the prediction pipeline and not meant as a physics statement or the best / most recent predictions from each of the generators for the given data set.

The right-hand side of Fig. 1 presents MARLEY predictions of flux-averaged differential cross sections for $\nu_e$ produced from $\mu^+$ decay at rest scattering on $^{40}$Ar. The blue histogram shows the inclusive charged-current prediction, and the other histograms show contributions arising from several distinct final states. The cross-section predictions obtained using NuHepMC without any MARLEY-specific code are very similar to those shown in a previous MARLEY publication [29]. An experimental measurement of this interaction channel at the relevant tens-of-MeV $\nu_e$ energy scale is not yet available, but adding future data points to the plot would be easily achieved with the present NuHepMC-based workflow.

Comparisons similar to those above can be easily produced for an arbitrary interaction channel using information available in the NuHepMC event record. A Monte Carlo estimator for the flux-averaged differential cross section in a bin of an arbitrary kinematic variable $x \in [x_k, x_{k+1})$ is given by

$$\left\langle \frac{d\sigma}{dx} \right\rangle_k \approx \frac{\langle \sigma \rangle \sum_{e=1}^{n} \delta_e^k w_e}{\Delta x_k \sum_{e=1}^{n} w_e} \tag{14}$$

where $\langle \sigma \rangle$ is the flux-averaged total cross section (see Sec. 3), $w_e$ is the statistical weight of the $e$-th event, $n$ is the number of generated events, and $\Delta x_k = x_{k+1} - x_k$ is the width of the $k$-th bin. The symbol $\delta_e^k$ evaluates to unity when the value of $x$ from the $e$-th event falls within

the $k$-th bin and zero otherwise. In the case of unit weight events, the value of $w_e = 1$ for all events, and $\sum_{e=1}^{n} w_e = n$. One thus recovers the traditional definition of these estimators for unweighted events.

The Monte Carlo statistical uncertainty (standard deviation) on the estimator of the differential cross section is approximately given by

$$\text{StdDev}\left(\left\langle \frac{d\sigma}{dx} \right\rangle_k\right) \approx \frac{\langle \sigma \rangle}{\Delta x_k \sum_{e=1}^{n} w_e} \sqrt{\sum_{e=1}^{n} \delta_e^k w_e^2} \tag{15}$$

where we have assumed that $\langle \sigma \rangle$ is exactly known. For cases where it is estimated using Monte Carlo techniques rather than directly calculated, the statistical uncertainty discussed in B also applies.

## Acknowledgements

The authors would like to thank A. Papadopoulou for sharing a NUISANCE implementation for the example comparison in Sec. 4. This manuscript has been authored by Fermi Research Alliance, LLC under Contract No. DE-AC02-07CH11359 with the U.S. Department of Energy, Office of Science, Office of High Energy Physics. The work in this manuscript is supported by the Royal Society, grant number URF\R1\211661.

## A  Target Scaling Conventions

It is useful to illustrate the relationship between the different cross-section target scaling conventions discussed in G.C.4 with an example. Let $\sigma^C$ and $\sigma^H$ respectively denote the total cross section for a given process to occur with a $^{12}$C and $^1$H atom. Then the cross section for an interaction with the carbon part and the hydrogen part of a $CH_2$ molecule under the different target scaling conventions are given below:

- "PerTargetAtom":

  - Carbon interaction: $\sigma = \sigma^C$
  - Hydrogen interaction: $\sigma = \sigma^H$

- "PerTargetMolecule":

  - Carbon interaction: $\sigma = \sigma^C + 2\sigma^H$
  - Hydrogen interaction: $\sigma = \sigma^C + 2\sigma^H$

- "PerTargetNucleon":

  - Carbon interaction: $\sigma = \frac{1}{12}\sigma^C$
  - Hydrogen interaction: $\sigma = \sigma^H$

- "PerTargetMolecularNucleon":

  - Carbon interaction: $\sigma = \frac{1}{14}(\sigma^C + 2\sigma^H)$
  - Hydrogen interaction: $\sigma = \frac{1}{14}(\sigma^C + 2\sigma^H)$

It should be clear from this example why we recommend against implementations using `"PerTargetMolecularNucleon"`, important information about the relative cross section for the nuclear constituents is lost when averaging over the molecule—the cross-section information reported for an interaction with the hydrogen part of a hydrocarbon will implicitly contain carbon nuclear effects. However, the scheme is discussed here because where cross-section information is written out at the event level in existing neutrino event generators, it is often used. It is usually possible to reconstruct the cross-section for constituents by brute force from a sample of events.

## B Estimation via Monte Carlo sampling

In this section we provide two example methods of estimating the flux-averaged total cross-section when it is not known *a priori*. The first method is suitable for GENIE-like codes that calculate energy-dependent total cross sections prior to event generation itself. The second is suitable for ACHILLES and similar codes that do not use precalculated cross sections in this way.

### B.1 GENIE-like approach

The (anti)neutrino interactions that occur in the volume $V$ are drawn from the probability distribution

$$P(f, j, E_\nu, \vartheta_\nu, \varphi_\nu, \mathbf{x}, t) = \frac{1}{N} \phi \rho \sigma. \tag{B.1}$$

It follows from these definitions that

$$\langle \sigma \rangle = \left[ \sum_f \sum_j \int P(f, j, E_\nu, \vartheta_\nu, \varphi_\nu, \mathbf{x}, t) \frac{1}{\sigma} dE_\nu d\cos\vartheta_\nu d\varphi_\nu d^3\mathbf{x} dt \right]^{-1}. \tag{B.2}$$

One may therefore obtain a Monte Carlo estimator $\hat{\sigma}$ for the flux-averaged total cross-section $\langle \sigma \rangle$ via the expression

$$\hat{\sigma} = \left[ \frac{\sum_{e=1}^{n} \frac{w_e}{\sigma_e(f, j, E_\nu)}}{\sum_{e=1}^{n} w_e} \right]^{-1} \tag{B.3}$$

where $w_e$ is the statistical weight for the $e$-th event, $n$ is the total number of generated events, and $\sigma_e$ is the total cross section evaluated for the (anti)neutrino species ($f$), target ($j$), and incident energy ($E_\nu$) sampled in the $e$-th event. An estimator for the Monte Carlo statistical standard deviation of $\hat{\sigma}$ is given by

$$\text{StdDev}(\hat{\sigma}) = \sqrt{\text{Var}(\hat{\sigma})} = \hat{\sigma}^2 \cdot \sqrt{\text{Var}\left(\frac{1}{\hat{\sigma}}\right)} = \hat{\sigma}^2 \cdot \sqrt{\frac{n \cdot \sum_{e=1}^{n} w_e^2 \left(\frac{1}{\sigma_e} - \frac{1}{\hat{\sigma}}\right)^2}{\left(\sum_{e=1}^{n} w_e\right)^2 \cdot (n-1)}}. \tag{B.4}$$

Note that there is no universally accepted expression for the standard error on a weighted arithmetic mean such as the one computed in Eq. B.3 (before taking the reciprocal of the expression). The choice used in Eq. B.4 is based on the third expression for "SEM$_\text{w}$" recommended in Ref. [30] based on bootstrapping studies (see also references therein). The quantity proposed as an estimator for the square of the standard error on a weighted mean

$$\bar{x}_w = \frac{\sum_{e=1}^{n} w_e x_e}{\sum_{e=1}^{n} w_e} \tag{B.5}$$

of $n$ values $x_e$ is

$$\text{Var}(\bar{x}_w) = \frac{n}{(n-1)(n\bar{w})^2} \left[ \sum_{e=1}^{n} (w_e x_e - \bar{w}\bar{x}_w)^2 - 2\bar{x}_w(w_e - \bar{w})(w_e x_e - \bar{w}\bar{x}_w) + \bar{x}_w^2(w_e - \bar{w})^2 \right]$$

(B.6)

where

$$\bar{w} = \frac{1}{n} \sum_{e=1}^{n} w_e.$$

(B.7)

The expression in Eq. B.6 may be simplified to read

$$\text{Var}(\bar{x}_w) = \frac{n}{(n-1)\left(\sum_{e=1}^{n} w_e\right)^2} \sum_{e=1}^{n} w_e^2 (x_e - \bar{x}_w)^2.$$

(B.8)

The result in Eq. B.4 is obtained immediately with the substitutions $x_e \to 1/\sigma_e$ and $\bar{x}_w \to 1/\hat{\sigma}$.

While the expressions in Eqs. B.3 and B.4 may be useful for estimating $\langle \sigma \rangle$ from a sample of generated events, they require access to the entire sample and are thus unsuitable for implementing the running estimate described in E.C.4. However, an adaptation of West's algorithm [31] provides a solution for this application. Let

$$n_0 = S_0 = \mu_0 = T_0 = 0.$$

(B.9)

Then, for the $e$-th event, let the values of these quantities be defined recursively via

$$n_e = n_{e-1} + 1 = e$$

(B.10)

$$S_e = S_{e-1} + w_e$$

(B.11)

$$\mu_e = \mu_{e-1} + \frac{w_e}{S_e} \left( \frac{1}{\sigma_e} - \mu_{e-1} \right)$$

(B.12)

$$T_e = T_{e-1} + w_e^2 \left( \frac{1}{\sigma_e} - \mu_{e-1} \right) \left( \frac{1}{\sigma_e} - \mu_e \right),$$

(B.13)

where $\sigma_e$ and $w_e$ are assigned the same meanings as above.

The running Monte Carlo estimator $\hat{\sigma}_e$ of $\langle \sigma \rangle$ for the $e$-th event ($e > 0$) may then be written as

$$\hat{\sigma}_e = \frac{1}{\mu_e}.$$

(B.14)

Its estimated statistical uncertainty is given by

$$\text{StdDev}(\hat{\sigma}_e) = \frac{1}{\mu_e^2} \sqrt{\frac{n_e T_e}{(n_e - 1) S_e^2}}.$$

(B.15)

## B.2 ACHILLES-like approach

When calculating Eq. (4), the equation can be expanded to include the integrals over the differential cross section giving

$$N = \sum_f \sum_j \int dE_\nu \, d\cos\vartheta_\nu \, d\varphi_\nu \, d^3\mathbf{x} \, dt \, d\Omega \, \phi_f \, \rho_j \, \frac{d\sigma_{fj}}{d\Omega},$$

(B.16)

where $\Omega$ is the final state phase space and the cross section is broken up into individual processes as $\sigma_{fj}$. The equation can then be estimated using traditional Monte-Carlo methods giving

$$N \approx \frac{V}{n} \sum_f \sum_j \sum_i \phi_f(x_i) \rho_j(x_i) \frac{d\sigma_{fj}}{d\Omega}(x_i),$$

(B.17)

where $x_i$ are the selection of the variables of integration for the $i$th point and $V$ is the volume of the space being integrated. The uncertainty on the integral estimate is thus given by the traditional calculation of the standard deviation. The variance of the integral estimate can be improved through the use of importance sampling, such as VEGAS [32, 33]. The integral over the neutrino fluxes and over the density of the nuclear targets can be obtained in a similar method, or by using other numerical integration techniques like quadrature. The flux-averaged cross section can thus be calculated with Monte-Carlo techniques using Eq. 8.

The results of the above Monte-Carlo calculation would produce a set of weighted events that can be used as is, or can be unweighted through the following procedure. First, the maximum values for each neutrino species and nuclei can be estimated using Monte-Carlo methods ($w_{fj}^{\max}$). Second, a neutrino species and nuclei is selected according to the probability

$$P_{fj} = \frac{w_{fj}^{\max}}{w^{\max}}, \tag{B.18}$$

where $w^{\max}$ is given by the sum of the maximum weight over all neutrino types and nuclei. Once a neutrino type and nucleus is selected, a set of initial and final state momenta are generated and the integrand for that particular neutrino type and nucleus is calculated. The event can then be unweighted by performing an accept-reject step using the ratio of weight of the event to the maximum weight calculated for the given neutrino type and nucleus. In other words, the weight of an event would be given by

$$\tilde{w}_i = w^{\max} \sum_{fj} \Theta\left(\frac{w_{fj}^{\max}}{w^{\max}} - \sum_{\substack{f'<f, \\ j'<j}} \frac{w_{f'j'}^{\max}}{w^{\max}} - R_1\right) \Theta\left(\frac{w_i}{w_{fj}^{\max}} - R_2\right), \tag{B.19}$$

with $R_1, R_2 \in [0, 1]$ a uniformly distributed random number. This would result in a collection of events with either weights $w^{\max}$ or zero, and the average of these events would give an estimate of the total flux-averaged cross section, and only the events with non-zero weight are required to be written out as long as the number of attempted events (i.e. the total including the zero weight events) is also tracked. There are technical issues with directly using the true maximum sampled, since the maximum depends on the number of samples taken. There are many ways to mitigate this numerical issue, one approach is discussed in Section 4A of Ref. [34].

## C  Example Event Graphs

Figs. 2, 3, and 4 show some example event graphs for neutrino event generators that natively implement, or are convertible-to, the NuHepMC format. Although the details of each generator's implementation of this standard differ, the constraints that are imposed enable consistent usage for the most common tasks without any knowledge of each generator's implementation details.

## D  Extracting Bibliography Information

If the NuHepMC file signals that convention G.C.6 has been used, then it is possible to run the file through an external tool called HEPREFERENCE [35][1] to produce a BibTeX file and a short

---

[1]The code can be found at: https://github.com/NuDevTools/HEPReference

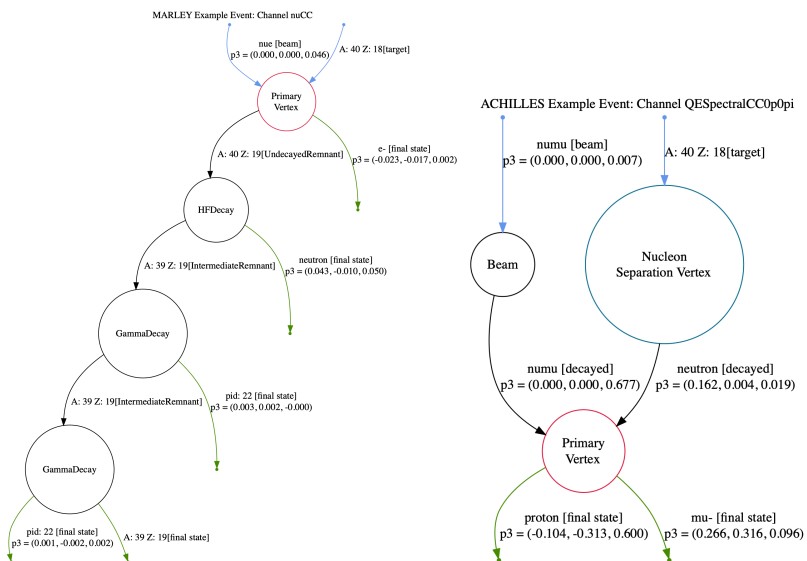

Figure 2: (left) A MARLEY event graph in the NuHepMC format. (right) An ACHILLES event graph in the NuHepMC format.

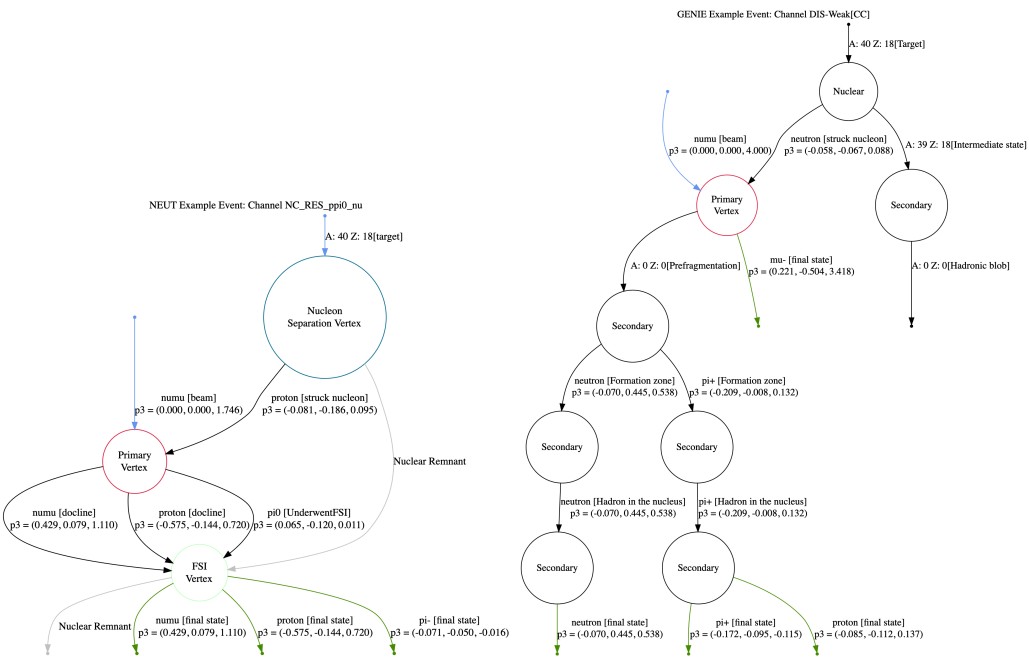

Figure 3: (left) A NEUT event graph in the NuHepMC format. (right) A GENIE event graph in the NuHepMC format.

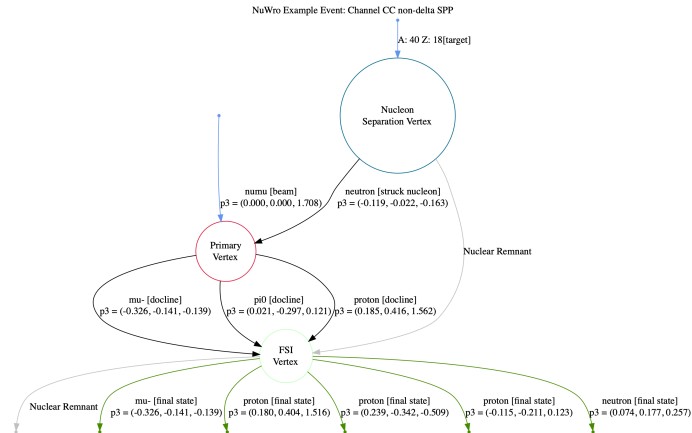

Figure 4: A NuWro event graph in the NuHepMC format.

text blurb showing how to cite all the papers used to produce the events. An example use case can be found in the repository.

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
