# Peer review of "NuHepMC: A standardized event record format for neutrino event generators"

_SciPost Physics Codebases, doi:SciPost Phys. Codebases 57-r1.0 (2025) , SciPost Phys. Codebases 57 (2025)_

## Round 2 · Referee Report · Anonymous (Referee 1) · 2024-9-11

Strengths

This manuscript presents a new format for the output of neutrino generators, aimed at establishing a community-wide standard. The authors argue that the format will facilitate comparisons across various event generators. The format builds on the structures developed for the HepMC3 event record library, which is already in use in other HEP-ex fields.

Weaknesses

The authors should address some points, listed in the "Requested changes."

Report

Overall, I find this to be a valuable contribution that deserves publication. However, I believe the authors should address some points for clarity and completeness.

Requested changes

General comments:

Repository: It would be highly beneficial to include a link to a repository. Specifically, I recommend that the authors provide the scripts used to generate Figure 1. This will greatly enhance the utility of the manuscript for readers.

Dependencies: The authors should clarify the dependencies that the proposed format introduces for event generators. For instance, does this imply that generators need to install HepMC3? Does this limit the format to C++-based generators, or are there options for Python-based codes? Are there Python bindings for the NuHepMC structures? A detailed explanation of how to integrate this format into existing generators and any associated limitations would be helpful.

Specific comments:

  • Page 8: The attributes for "Beam Energy Distribution Description" are not well-suited for atmospheric neutrino experiments, which involve multiple energies and baselines. The manuscript should clarify how such information can be incorporated.

  • Page 9: The term "Nevents" should be clarified—does it refer to the number of generated events or the number of interactions?

  • Page 9: Currently, "ParticleStatusInfo" is labeled as a "Suggestion". Why not formalize this as a "Convention"?

  • Page 9: It might be useful to include a label that informs users about the reference frame used for all the provided data.

  • Page 11: Table 1 lists several processes. The difference between SIS and DIS should be clearly explained for better understanding.

  • Page 13: The link between "outgoing real particle" and "observable" is unclear. For instance, particles like taus, which can decay but are observable at high energy, could fit this category. This distinction should be explained in more detail.

  • Page 21-22: Figures 2, 3, and 4 are difficult to read. Please enlarge these figures for better legibility.

Recommendation

Publish (meets expectations and criteria for this Journal)

  • validity: good
  • significance: good
  • originality: good
  • clarity: good
  • formatting: excellent
  • grammar: excellent

Author:  Steven Gardiner  on 2025-06-20  [id 5587]

(in reply to Report 1 on 2024-09-11)

We thank the referee for the review of the manuscript and the helpful comments. Replies and explanations of changes made before resubmission are given below.

Repository: It would be highly beneficial to include a link to a repository. Specifically, I recommend that the authors provide the scripts used to generate Figure 1. This will greatly enhance the utility of the manuscript for readers.

We thank the referee for this suggestion and agree that this addition would be useful. The location of the code to generate the right-hand plot from Figure 1 has now been referenced in the footnotes. We have also added a link in the main text to a supplemental repository (https://github.com/NuHepMC/cpputils) that provides example C++ scripts for manipulating events stored in the NuHepMC format. The MicroBooNE data set shown in Fig. 1 is available for use in automated NuHepMC-based comparisons within the NUISANCE software package.

Dependencies: The authors should clarify the dependencies that the proposed format introduces for event generators. For instance, does this imply that generators need to install HepMC3? Does this limit the format to C++-based generators, or are there options for Python-based codes? Are there Python bindings for the NuHepMC structures? A detailed explanation of how to integrate this format into existing generators and any associated limitations would be helpful.

We agree that this is an important clarification, and we have added some text to the paper accordingly.

Because NuHepMC represents a set of guidelines for using existing data structures from HepMC3 to represent neutrino scattering events, there are no extra dependencies required beyond anything needed to process generic HepMC3 events. Since the HepMC3 data format can be represented in simple ASCII text files, an event generator may produce output compliant with the NuHepMC standard without recourse to any external code whatsoever. Exactly this approach was used in draft code provided by the authors for proposed inclusion in a release of the GiBUU event generator; the NuHepMC output implementation was written in regular Fortran with no external dependency.

That being said, the HepMC3 reference library provides a standard implementation of the event format with many convenient tools that facilitate reading/writing the metadata recommended in the NuHepMC standard. We have ourselves created some example NuHepMC tools (C++ and Python) that themselves depend on the reference library. In particular, more advanced applications of NuHepMC, such as interoperability between multiple event generators, will likely be much simpler to implement by relying on the reference HepMC3 library.

  • Page 8: The attributes for "Beam Energy Distribution Description" are not well-suited for atmospheric neutrino experiments, which involve multiple energies and baselines. The manuscript should clarify how such information can be incorporated.

We agree with the referee that these attributes are not well-suited for atmospheric neutrino experiments. Standardizing representations of general neutrino fluxes is a complex topic that we leave to future work, but we wanted to provide an initial convention for representing some relevant information in the specific context of the accelerator neutrino community.

We have therefore added text to the paper that highlights that this is a preliminary definition of how to store flux metadata into the NuHepMC event record. We emphasize the need for additional work on standardizing flux formats across multiple kinds of neutrino experiments.

  • Page 9: The term "Nevents" should be clarified—does it refer to the number of generated events or the number of interactions?

We thank the referee for pointing out the need for greater clarity on this point. The "Nevents" attribute was intended to assist users in reproducing the events from a particular event generator run; we consider its exact interpretation to be an implementation detail. However, we have tidied up this section, and this example no longer appears in the manuscript.

  • Page 9: Currently, "ParticleStatusInfo" is labeled as a "Suggestion". Why not formalize this as a "Convention"?

We believe that the referee is referring to G.S.2 here, but we are not completely sure. Note that event generators are required (originally submitted G.R.6, now G.R.10) to provide definitions of all particle status codes used beyond those already standardized in HepMC3. The suggestion G.S.2 only adds explicit definitions of the HepMC3 codes as well. Since an unambiguous interpretation of the output is possible without this addition, for now we prefer to leave the inclusion of the standard codes as only a suggestion.

  • Page 9: It might be useful to include a label that informs users about the reference frame used for all the provided data.

There is a de facto standard among event generators that the lab frame is used to record particle momenta and vertex positions. This choice is also most convenient for use in experimental production workflows and in comparing predictions to cross-section measurements. We therefore adopt the lab frame in keeping with standard practice in the field.

However, we believe all information needed to transform to other reference frames can be calculated from the 4-vectors stored in the event. If it is desirable to report observables in other frames in the context of a specific model, these may be added as metadata to relevant particles and vertices as appropriate. We have added some text to the manuscript to discuss this issue in light of this comment from the referee. In particular, a new requirement (E.R.9) makes the choice of the lab frame explicit.

  • Page 11: Table 1 lists several processes. The difference between SIS and DIS should be clearly explained for better understanding.

We agree that the boundary between SIS and DIS is ill defined, and we have updated the text to add an explanation. Ultimately we emphasize that these process ID ranges represent rough guidelines for generator authors rather than strict definitions.

  • Page 13: The link between "outgoing real particle" and "observable" is unclear. For instance, particles like taus, which can decay but are observable at high energy, could fit this category. This distinction should be explained in more detail.

We initially intended these phrases to be synonyms in this context. To improve clarity, we have decided to simply remove the relevant sentence.

  • Page 21-22: Figures 2, 3, and 4 are difficult to read. Please enlarge these figures for better legibility.

The figures have been enlarged as requested.

---

## Round 2 · Referee Report · Anonymous (Referee 2) · 2024-10-7

Strengths

The authors have developed and documented a valuable tool for the study of neutrino interactions with matter using the various existing event generators, setting a standard for future developments.

Weaknesses

I have only a number of minor comments and questions to be addressed. See below.

Report

I am in favor of the publication in SciPost.

Requested changes

- To deal with the fact that only in some cases the averaged cross section is known at the beginning of the simulation, the mixed solution of G.C.5 and E.C.4 is proposed. However, the association of unrealistic <sigma> to events generated early in the run could lead to confusion and mishandling among the users. I wonder if <sigma> could be stored by a procedure analogous to G.C.5 but at the end of the run, rather than at the beginning. Probably, there are reasons why a solution of this kind is not possible/feasible/convenient but perhaps a short explanation, not just for this referee but for the readers and potential users would be useful.
- In G.R.8, the authors refer to “generator-specific quasiparticles”, presumably with unresolved nuclear remnants in mind. However, the name “quasiparticle” has a different meaning in physics, which could lead to misunderstanding. It is therefore advisable to find a different name. For example, if no other “quasiparticles” than nuclear remnants are expected, the later term could be directly used.
- In connection to the previous point, in P.C.2, “the new particle number 2009900000 to correspond to a nuclear remnant pseudo-particle” is defined. On the other hand, in a given simulation there might be more than one kind of such remnants, for instance when there are different targets. Wouldn’t it then be convenient to have a range rather than a single number associated to these states?
- The authors strongly recommended adopting picobarns as cross section unit but they are certainly aware that its use is practically inexistent in the neutrino cross section literature.
- In table 1, ID ranges are associated with process categories. The latter correspond to the common classification adopted in the field but it is not necessarily a good choice because it is model dependent and, in some cases, there is no consensus about the kinematic boundaries of each region. Furthermore, not all generators provide information in these terms in their output. In my opinion it would be wise to be able to redefine this structure in the future.
- The beginning of Sec. 4 refers to “preliminary tools for converting proprietary neutrino event formats to the NuHepMC standard”. Such tools would be highly valuable for potential users. I understand the authors are not yet in the position to include these tools in this release but a sentence promising them (soon) would be encouraging.
- In connection to the MARLEY prediction in Fig 1 (right) it is stated that “adding future data points to the plot would be easily achieved with the present NuHepMC-based workflow.” It is not clear in which way the new format would help in this comparison. I would naively say that once the results of the simulation have been obtained, adding (future) data to the plot should be straightforward no matter the format.
- Finally, with the font chosen for Fig. 1, characters “I” and “1” look strange.

Recommendation

Ask for minor revision

---

## Round 3 · Author Response

List of changes
- Many clarifications were added based on reviewer feedback. Some topics addressed include the degree of dependence on the HepMC3 reference library, the use of the laboratory frame for quantities stored in the event record (see new requirement E.R.9), the interpretation of our recommended numerical ranges for process IDs, and the use of our proposed "catch-all" particle code for nuclear remnants that are not fully simulated.
- The description of target scaling was simplified. A related appendix was removed since it is no longer necessary.
- Several conventions were promoted to requirements, notably two related to the flux-averaged total cross section.
- A suggestion was added that provides a format for reporting the atomic abundances in a composite target material.
- A table giving the current state of NuHepMC adoption among the most widely-used neutrino event generators was added.
- Various small fixes were made, and some attributes were renamed.

---

## Round 3 · List of Changes

- Many clarifications were added based on reviewer feedback. Some topics addressed include the degree of dependence on the HepMC3 reference library, the use of the laboratory frame for quantities stored in the event record (see new requirement E.R.9), the interpretation of our recommended numerical ranges for process IDs, and the use of our proposed "catch-all" particle code for nuclear remnants that are not fully simulated.
- The description of target scaling was simplified. A related appendix was removed since it is no longer necessary.
- Several conventions were promoted to requirements, notably two related to the flux-averaged total cross section.
- A suggestion was added that provides a format for reporting the atomic abundances in a composite target material.
- A table giving the current state of NuHepMC adoption among the most widely-used neutrino event generators was added.
- Various small fixes were made, and some attributes were renamed.

---

## Editorial Decision

published